# Cerebral Vein Thrombosis in the Antiphospholipid Syndrome: Analysis of a Series of 27 Patients and Review of the Literature

**DOI:** 10.3390/brainsci11121641

**Published:** 2021-12-13

**Authors:** Alba Jerez-Lienas, Alexis Mathian, Jenifer Aboab, Isabelle Crassard, Miguel Hie, Fleur Cohen-Aubart, Julien Haroche, Denis Wahl, Ricard Cervera, Zahir Amoura

**Affiliations:** 1Department of Autoimmune Diseases, Universitat de Barcelona, Hospital Clínic, 08036 Barcelona, Spain; ajerez@mutuaterrassa.cat; 2Systemic Autoimmune Diseases Unit, Department of Internal Medicine, Universitat de Barcelona, Hospital Universitari Mútua de Terrassa, 08221 Terrassa, Spain; 3French National Referral Center for Systemic Lupus Erythematosus, Antiphospholipid Antibody Syndrome and Other Autoimmune Disorders, Service de Médecine Interne 2, Institut E3M, Inserm UMRS, Centre d’Immunologie et des Maladies Infectieuses (CIMI-Paris), Assistance Publique–Hôpitaux de Paris (APHP), Groupement Hospitalier Pitié–Salpêtrière, Sorbonne Université, 75013 Paris, France; alexis.mathian@aphp.fr (A.M.); jenifer.aboab@aphp.fr (J.A.); miguel.hie@aphp.fr (M.H.); fleur.cohen@aphp.fr (F.C.-A.); julien.haroche@aphp.fr (J.H.); zahir.amoura@aphp.fr (Z.A.); 4Department of Neurology, Hôpital Lariboisière, 75010 Paris, France; isabelle.crassard@lrb.aphp.fr; 5Department of Vascular Medicine, Referral Center for Rare Vascular Diseases, Centre Hospitalier Régional Universitaire de Nancy, 54500 Vandoeuvre-lès-Nancy, France; d.wahl@chru-nancy.fr

**Keywords:** antiphospholipid syndrome, antiphospholipid antibodies, cerebral vein thrombosis, intracranial sinus thrombosis

## Abstract

(1) Background: The Antiphospholipid Syndrome (APS) is a systemic autoimmune disorder characterized by arterial and/or venous thrombosis, pregnancy morbidity and raised titers of antiphospholipid antibodies. Cerebral vein thrombosis (CVT) is a rare form of cerebrovascular accident and an uncommon APS manifestation; the information in the literature about this feature consists of case reports and small case series. Our purpose is to describe the particular characteristics of CVT when occurs as part of the APS and compare our series with the patients published in the literature. (2) Methods: We conducted a retrospective observational study collecting data from medical records in three referral centers for APS and CVT, and a systematic review of the literature for CVT cases in APS patients. (3) Results: Twenty-seven APS patients with CVT were identified in our medical records, the majority of them diagnosed as primary APS and with the CVT being the first manifestation of the disease; additional risk factors for thrombosis were identified. The review of the literature yielded 86 cases, with similar characteristics as those of our retrospective series. (4) Conclusions: To our knowledge, our study is the largest CVT series in APS patients published to date, providing a unique point of view in this rare thrombotic manifestation.

## 1. Introduction

The antiphospholipid syndrome (APS) is a systemic autoimmune disorder characterized by arterial and/or venous thromboses, pregnancy morbidity and raised titers of antiphospholipid antibodies (aPL) [1,2]. It is considered to be the most common acquired thrombophilia. Thrombotic events in these patients consist in deep vein thrombosis (32%), stroke (13%), superficial thrombophlebitis (9%), pulmonary embolism (9%) and transient ischemic attack (7%), among other clinical manifestations. Stroke is the most frequent arterial thrombosis [3].

Cerebral vein thrombosis (CVT) is the thrombotic occlusion of a cerebral sinus or parenchymal vein, which occurs predominantly in young women (median age, 39 years; female/male ratio, 3/1) [4]. It is a rare disease, as it represents only 0.5–1% of the cerebral vascular accidents [5] and has an estimated incidence of 1.32 per 100.000 person-years [6].

The clinical picture in a patient presenting with CVT will differ according to the location of the thrombosis. When it affects a cerebral parenchymal vein, it induces a cytotoxic and vasogenic edema that is more prominent than in an arterial stroke, and it can evolve into a hemorrhagic stroke due to high pressure in the veins. If the thrombosis affects a cerebral sinus vein, it diminishes the cerebrospinal fluid absorption, which may lead to intracranial hypertension [7].

Many risk factors and causes have been related to the development of CVT; however, in 12.5% of adult patients with CVT no cause is identified after a thorough search [4]. Conversely, in more than half of patients with CVT more than one risk factor is identified, which encourages a methodical clinical search for risk factors, even in those patients who have an obvious cause [4], as it may modify the prognosis and clinical management. The CVT etiology has changed in developed countries since the development of antibiotics, as it used to be mainly due to an infectious disease [4,8,9,10], but currently thrombophilia (either genetic or acquired) and oral contraceptives seem to be the most common risk factors. Nowadays, thanks to the advanced imaging techniques, more cases of CVT are diagnosed, even in the setting of paucisymptomatic patients, expanding our knowledge on associated causes. The use of anticoagulant treatments has also improved its prognosis.

Currently, the most frequent etiology of CVT is genetic thrombophilia, which can be identified in 20–30% of patients. The most commonly found thrombophilia are factor V Leiden mutation (15–17%) and prothrombin gene mutation *G20210A* (10–12%) [11]. In a series of 121 patients with CVT, hyperhomocystinemia was identified in 20% of the cases [12]. In female patients, oral contraceptives are a prominent cause, frequently associated with other risk factors [13].

As for systemic inflammatory diseases, in the International Study on Cerebral Vein and Dural Sinus Thrombosis (ISCVT) series of 624 patients [4], the underlying cause was APS in 5.9% of patients, inflammatory bowel disease in 1.6%, Behçet disease in 1% and systemic lupus erythematosus (SLE) in 1% of cases. In the 1000 APS patient’s series of Cervera et al. [3], only 7 patients had a CVT. Therefore, CVT is an uncommon manifestation in APS patients, which makes it difficult to collect medical information in order to expand our knowledge in this topic; the information available in the literature is scarce and most of the published studies are case reports or short case series.

On this study, we conducted a retrospective observational study in three university care referral center hospitals in order to describe the clinical features, diagnostic procedures, therapeutic options, outcome and follow-up characteristics of APS patients with CVT. We also conducted a review of the literature to gather the clinical features, diagnostic procedures, therapeutic options, outcome and follow-up characteristics of the CVT cases in APS patients published in the literature, and compared them to our retrospective series, in order to evaluate the homogeneity of their clinical characteristics.

## 2. Materials and Methods

We conducted a retrospective study in three main referral hospitals, collecting data from patients with CVT associated to APS: Service de Médecine Interne 2 of Hôpital Pitié-Salpêtrière, Paris, France (French national referral center for SLE and APS patients); the Department of Neurology of Hôpital Lariboisière, Paris, France (referral center for rare vascular diseases of the brain and the eye (CERVCO)) and the Department of Autoimmune Diseases of Hospital Clínic, Barcelona, Catalonia, Spain (referral center for autoimmune diseases). The search for clinical cases was made through review of the medical database of each department, from January 1986 to July 2016 in the Service de Médecine Interne 2 Hôpital Pitié-Salpêtrière, from September 1997 to June 2013 in the Department of Neurology of Hôpital Lariboisière and from January 1985 to September 2021 in the Department of Autoimmune Diseases of Hospital Clínic.

We included patients with CVT in association with APS, according to the following criteria: The diagnosis of CVT was based on clinical suspicion due to a compatible clinical picture and confirmed with a brain imaging technique (computed tomography scan (CT scan) or magnetic resonance imaging (MRI), with or without intravenous contrast) with typical abnormalities [14,15]. In all cases, the diagnosis of CVT was confirmed by a neuroradiologist.

The diagnosis of APS was based on the available criteria at the time of the patient evaluation in the referral hospital. Accordingly, patients diagnosed with APS prior to 2006 fulfilled the Sapporo criteria [16], while subsequent cases fulfilled the Sydney criteria for APS [17]. Besides the thromboembolic event of the CVT, patients had to present the following laboratory criteria:Sapporo criteria (before 2006): two positive results determined 6 weeks apart for anticardiolipin antibodies (aCL) at high or medium titer (or >99th percentile), IgG or IgM isotypes, or the presence of circulating lupus anticoagulant (LA).Sydney criteria (after 2006): two positive results, 12 weeks apart, for aCL, IgG or IgM isotypes, at titer > 40 GPL/MPL or >99th percentile; or the presence of anti-beta-2-glycoprotein antibodies (aß2GPI) at >99th percentile; or the presence of circulating lupus anticoagulant (LA).All patients underwent a laboratory search for aPL on their referral hospital, and LA presence was confirmed by a second test, as recommended by the international guidelines.

According to the APS criteria, we excluded all patients with CVT whose laboratory findings suggestive of APS were not available for 5 years after the thrombotic event.

A systematic review of the literature was performed in order to obtain all available information on CVT and APS, as well as to collect all cases of CVT in association to APS. The search for relevant articles was conducted using Medline on the PubMed web page, with the following keywords (MeSH terms) composed formula: [“Antiphospholipid Syndrome” OR “Antibodies, Antiphospholipid”] AND “Intracranial Thrombosis”. In addition, some articles were included by review of references from selected articles and by manual search. Last access to the web was on September 2021. We selected articles in English, French, Spanish and German, from January 1983 to September 2021.

Articles were selected if their main topic was CVT, APS or the association of both conditions, including reported cases of patients with CVT associated to APS. To be included in the case series of the literature, the reported clinical cases had to fulfill the same inclusion criteria applied to our case series, as described above. Cases lacking the following information in the article description were excluded: APS classification, previous APS manifestations, clinical signs and symptoms leading to the diagnosis of the CVT, aPL profile, CVT location, as well as associated intra- or extra-parenchymal complications, treatment and outcome of the acute injury. Clinical cases of patients younger than 13 years of age were excluded.

All clinical data from the patients in our retrospective series were collected from the medical reports. At the time of CVT, the following data were gathered: epidemiologic information (age, gender), clinical picture, imaging technique leading to the diagnosis, CVT location and extension (CVT was considered extended if affecting more than one sinus vein), presence of intra- or extra-parenchymal complications (venous infarction or intracranial bleeding), presence of intracranial hypertension (diagnosed by observation of papilledema on fundoscopy), treatment given on the acute phase and duration of the hospital admission. The diagnostic delay was determined by the time between the onset of the symptoms and the CVT diagnosis. Depending on the diagnostic delay, three groups of clinical presentation were established: the clinical presentation was considered acute when the diagnostic delay was less than 48 h, subacute when the diagnostic was made between two days and one month after the onset of symptoms and chronic if the diagnostic delay was superior to one month. Details on the anticoagulant treatment were also collected, including type of treatment, duration and the targeted International Normalized Ratio (INR).

CVT have been related to many risk factors [4], whose presence was registered on our case series whenever affected a patient concomitant to the CVT. We also gathered information on the thrombophilia tests conducted on each case, usually: factor V Leiden mutation (*G1691A*), factor II mutation (*G20210A*), protein C, protein S and antithrombin III deficiencies and homocysteine levels. Anemia was considered a risk factor when the hemoglobin levels were <10 g/dL.

Regarding APS, clinical information was gathered on criteria and non-criteria APS manifestations according to Sapporo and Sydney classifications [16,17,18,19] and on thrombotic and obstetric manifestations that took place before, simultaneously or after the CVT. All information was obtained from medical reports.

The aPL profile of every patient was registered carefully: isotype, titer and the standard used by the laboratory. Patients were classified as having primary APS or APS associated to another disease. SLE was defined by the classification criteria that were available at the moment [20,21].

We gathered medical information on the outcomes and follow-up, taking into account the type of sequelae if present, and assessing the degree of neurologic handicap with the modified Rankin Scale (mRS) [22] at hospital discharge, after 6 months and after one year (Table 1).

According to the mRS a year after the CVT episode, patients were classified into three prognostic groups: complete recovery (mRS 0–1), partial recovery with maintained autonomy (mRS 2), dependency (mRS 3–5) and deceased (mRS 6). After reviewing follow-up medical reports, data was also collected on APS manifestations, SLE manifestations and pregnancy.

Data from our descriptive retrospective observational series were analyzed in terms of percentage, mean and range. The statistical differences between our series and the cases from the literature were tested with the Mann–Whitney test (quantitative variables) and the Khi-2 test (qualitative variables). All tests were bilateral and took into account a *p* value <0.05. The statistical analysis was performed using SPSS software.

## 3. Results

### 3.1. Cerebral Vein Thrombosis

Twenty-seven patients were included in our retrospective series: 14 from Hôpital Pitié-Salpêtrière, 10 from Hôpital Lariboisière, and 3 from Hospital Clínic. The first case of CVT registered took place in December 1985. Most of them were women (22 patients; female/male ratio, 4.4/1) with a median age of 35.7 years (range 16–70) at the time of the CVT episode. The main characteristics of the CVT are described in Table 2.

Taking into account the time between the onset of symptoms and the moment when the diagnosis of CVT was made, the majority of patients had a subacute clinical presentation (15 patients), on four cases the clinical picture was considered chronic and on three cases, acute. There was no information on 5 patients on the medical reports concerning the onset of the clinical picture.

The most common symptom at presentation of the CVT was headache, affecting 24 of the 27 patients (88.9%) and it was the only symptom in 11 cases (40.7%). A focal neurologic deficiency was noted on 8 patients (29.6%): four motor deficiencies, 3 sensitive deficiencies, 3 patients with aphasia and one patient with cerebellum symptoms. Six patients had epileptic seizures, one of them went into epileptic status at admission. Three patients had and impaired level of consciousness (2 of them with Glasgow Coma Scale below 9 points). In addition, drowsiness was remarked in 2 patients and another 2 patients showed signs of confusion. Fundoscopy was performed in 10 patients, identifying signs of intracranial hypertension in three of them.

To diagnose CVT, a combination of imaging techniques was performed in half of patients (12 out of 25) including both CT scan (with or without iv contrast) and MRI (with or without angiography). There was a lack of information on this procedure in two patients. Four patients were diagnosed by MRI alone, CT scan with iv contrast alone was performed in 4 patients. An extended CVT was identified in 15 patients (57.7%), whereas 9 patients had thrombosis in an isolated venous sinus, two patients had only involvement of the cortical veins and in one case the location of the CVT was not recorded. The most frequently thrombosed venous sinus was the left transverse sinus (16 cases, 59%), followed by the superior sagittal sinus (10 cases, 37%), the right transverse sinus (9 cases, 33%), the left and right sigmoid sinuses (4 and 3 cases, 15% and 11%, respectively) and the straight sinus (3 cases, 11%). On 8 patients, the thrombosis was observed to involve the jugular vein as well. Furthermore, over half of the patients had intra- or extra-parenchymal complications (15 patients, 57.7%): bleeding was found in 9 patients (6 parenchymal hemorrhages, 3 subarachnoid hemorrhages) and six patients had venous infarction (two of them were hemorrhagic). Complications observed on the central nervous system imaging techniques were bilateral in seven cases.

Another risk factor for CVT was identified in 18 patients (66.7%); the most frequent one was the use of oral contraceptives (8 patients), followed by local infections (6 patients). Eighteen patients (66.7%) had at least one additional risk factor besides APS and seven patients (25.9%) had two or more risk factors besides APS.

Almost every patient was treated with low molecular weight heparin (LMWH) or unfractioned heparin (UFH) initially, switching to vitamin K antagonists (VKA) after a few days; only one patient was treated with corticosteroids (in 1985). No patients were treated with immunosuppressive drugs. In addition, two patients received acetazolamide for intracranial hypertension, two patients received antiepileptic drugs and two patients were treated with intravenous immunoglobulins due to thrombocytopenia (both of them had primary APS). When VKA were initiated, the targeted INR was between 2 and 3 in 14 patients (between 2.5 and 3 for two of them) and between 2.5 and 3.5 in 9 patients (between 3 and 3.5 in seven cases). The information on the targeted INR was missing in three patients.

Mean time until hospital discharge was 15.5 days (range 0–57 days). At discharge, the mRS was 0 in 6 patients, 1 in 18 patients and 4 in 2 patients (mean mRS, 1). No patient died. Grouped in three categories, most of the patients (24) had a good clinical status at discharge with a mRS of 0–1, no patients had a mSR of 2 and two patients were impaired for normal activities with mRS 3–5.

### 3.2. Antiphospholipid Syndrome

In our series, eighteen patients had primary APS, whereas APS was associated to SLE in 6 patients (one of them overlapped with systemic sclerosis), to essential thrombocytosis in 2 patients and to Sjögren’s syndrome in 1 patient. At the time of the CVT, one patient had a SLE flare. Regarding the classification criteria, 9 patients were diagnosed with Sapporo criteria. Reviewing the antibodies profile performed, all patients diagnosed under Sapporo criteria fulfilled the Sydney criteria.

A cerebral vein thrombosis was the first manifestation of APS in 20 out of 27 patients in our series (74.07%). At the time of the CVT episode, only two patients presented with a concomitant thrombosis in another venous territory: a deep vein thrombosis (DVT) and a pulmonary thromboembolism. Seven patients had non-criteria manifestations previous to the CVT or simultaneously: 4 had thrombocytopenia, 2 had mitral valvulopathy and one had livedo racemosa. None of the registered patients in our series presented as a catastrophic APS. The characteristics of the APS in our series are described on Table 3.

### 3.3. Follow Up

Mean time of follow-up was 66.8 months (range, 1–289 months). Anticoagulant treatment with VKA was maintained during follow-up in 21 patients, although in one case it was stopped after 6 years due to repeated falls. VKA were given in five patients during six months and then stopped and one patient did not receive anticoagulant treatment until 8 years after the CVT episode, when he presented DVT.

Main characteristics of follow-up and outcomes are described in Table 4. During follow-up, many patients underwent neuroimaging control tests and detailed information was available in 20 patients: ten of them had a partial repermeabilisation of the CVT, complete repermeabilisation was observed on 7 patients and the cerebral vein occlusion was persistent in 3 patients. None of the patients had a new CVT during follow-up.

Over half of the patients (15) suffered from diverse sequelae due to the CVT, the most common problem being headaches of variable intensity in ten cases. In six cases patients suffered from seizures, both focal and generalized. Two patients complained of cognitive disturbances, one developed a dural fistula type I that produced tinnitus and resolved spontaneously and one patient remained with hemiplegia and dysarthria (this patient suffered from diverse complications due to his neurological deficit, that lead to his death seven years after the CVT episode). At six months of follow-up, the mRS score was 0 in 9 patients, mRS 1 in 13 patients, mRS 2 in 1 patient and mRS 4 in 1 patient, with a mean mRS score of 0.79. It was not possible to obtain detailed information from the medical records in 3 patients. Grouped into three categories, most of the patients (22; 82%) had a good clinical status at 6 months with a mRS of 0–1, only one patient had a mSR of 2 and one patient was impaired for normal activities with mRS 3–5. After one year of follow-up, the mRS score was 0 in 16 patients, mRS 1 in 7 patients, mRS 2 in 1 patient and mRS 4 in 1 patient, with a mean mRS score of 0.59. It was not possible to obtain detailed information from the medical records in two patients. Grouped into three categories, most of the patients (23; 92%) had a good clinical status at one year with a mRS of 0–1, only one patient had a mSR of 2 and one patient was impaired for normal activities with mRS 3–5.

During follow-up, seven patients (28%) suffered new thrombosis and/or hemorrhages. There were 8 episodes of thrombosis in four patients (most of them DVT), one case of stroke and one transient ischemic attack. Three episodes happened when the patient was not taking VKA, three under VKA medication but with a low INR (below 2) and one case when the patient was taking only aspirin. Another patient had a thrombosis while taking VKA but there was no detailed information on the INR level at the time of the thrombosis. On the other side, five patients had hemorrhages: three were major bleedings [23] (one subarachnoid hemorrhage, one parenchymal bleeding and one psoas hematoma) and two were hematomas on the legs. Only one case had a documented high INR and in two patients detailed data was missing.

In our series, 14 of the 22 women were on a reproductive age. During the follow-up 5 pregnancies took place, from which four gave birth to healthy babies and only one had a stillbirth on second trimester.

### 3.4. Review of the Literature

Two hundred sixty-three articles were obtained after the Medline search. Fifteen were excluded due to language. Ninety-one articles were not related to APS or CVT, 79 articles were dismissed because of diverse reasons (pediatric cases, general reviews). In 29 articles there was not enough retrievable information or were not available. Finally, 14 articles were included through review of the references of the selected articles, for a total of 63 included articles. From those, 86 cases of CVT in APS patients were retrieved.

CVT cases in APS patients published in the literature affected predominantly young women (72% of women) with a mean age at the time of the CVT of 30.5 years (range 13–68). The most frequent forms of clinical presentation were acute (50%) and subacute (42%), with a mean diagnostic delay of 13 days (median 3, range 1–240). A chronic presentation was observed in 8% of cases. Headaches were the most frequent symptom, reported in 74% of published cases. Intracranial hypertension and the presence of a focal deficiency at admission had the same incidence (43%) and seizures (both focal and generalized) affected almost one third of the patients (33%). Twenty-one percent of the patients had an altered mental status and/or GCS < 9 at admission.

CVT was located in the lateral venous system (transverse sinuses, sigmoid sinuses) in 69% of patients, involved the superior sagittal sinus in 60% of patients and the straight sinus and deep venous system in 10% of patients. Cortical veins were thrombosed in 14% of cases. The majority of patients (60%) had an extensive thrombosis that involved more than one sinus.

By the time of hospital discharge, most patients had a complete resolution of the clinical picture or with mild symptoms remaining (mRS 0–1 in 73% of patients). Two patients had partial recovery with maintained autonomy (mRS score of 2 in 6% of patients) and 6 patients (11%) had a variable degree of disability (mRS 3–5). Five patients died during hospital admission (10%).

Besides APS, an additional risk factor was identified in over a half of the patients (64%): pregnancy or postpartum in 35% and oral contraceptives in 16% of the women in reproductive age, thrombophilia in 15% and only 5% of patients had a local infection.

Most patients had a primary APS (79%), only 13% were reported as APS associated to SLE and 7% associated to other diseases. Three patients had catastrophic APS.

Previous manifestations of APS were reported in 40% of patients: 21% of patients had had a previous thrombotic event, 4% had non-criteria manifestations and 32% of women in a reproductive age reported previous obstetric manifestations. No previous arterial thromboses were reported. The development of the CVT lead to APS diagnosis in the majority of patients (86%).

As for the aPL profile, 69% of patients had a single positive antibody, 26% were double positive and 5% were triple positive. The most frequently found antibody was aCL (66%), followed by LA (56%) and aβ_2_GPI (11%).

The detailed clinical characteristics of the CVT cases gathered from the literature are described in Table 5. We compare the characteristics of our series and the series of the literature on Table 6.

## 4. Discussion

The APS is an uncommon condition characterized by a hypercoagulability state that leads to the formation of clots, both in the arterial and venous territories. Nevertheless, the cerebral venous territory is a rare location for thrombosis in these patients, as demonstrated by its low incidence in larger APS series [3]. To our knowledge, our study represents the largest series of patients with APS and CVT.

In our series, CVT related to APS affects predominantly women at a young age (32 years), similar to observations made by Carhuapoma et al. [54] and Christopher et al. [87]. The clinical picture is homogeneus, generally with a subacute onset. Even though in the series from the literature an acute onset was more frequent (50%), the median diagnostic delay was of 3 days in the series from the literature and of 6 days in our retrospective series. Headache remained the most common symptom (89% of cases in our series, 74% in the series from the literature). No differences were found relating to the location of the thrombosis or the presence of CNS complications; an extensive sinus thrombosis was equally frequent. A single-center retrospective study conducted in China [88] reported similar results, with a retrospective series of 21 APS patients with CVT compared to matched non-APS controls with CVT. The APS patients were slightly younger (33 and 39 years old), but without statistical differences. Headache was the most common symptom (90.5%), but a chronic presentation was more frequent (52.4%). The CVT incidence in APS patients was higher than in other reports (7.8%), as well as mortality (15.8%), probably because it is a national referral center and more severe cases were received.

Interestingly, in our series, CVT was the first APS manifestation in almost three quarters of the patients (74%), leading to the diagnosis of the underlying APS (86% in cases from the literature). These data enlighten the importance of including the search for aPL when evaluating a case of CVT. But even when an obvious cause is evident (like a local infection), we recommend screening for thrombophilia in all cases, as in our series the majority of patients had more than one risk factor for the development of CVT (67%). Some previous studies have shown that the presence of additional risk factors increases the thrombotic risk in patients with APS [89,90,91]. In a prospective series of 404 APS patients, half of the patients had a combination of risk factors at the time of the first thrombosis. As for arterial thrombosis, in the RATIO study [92] patients with an ischemic stroke frequently had cardiovascular risk factors associated to the presence of aPL. These observations suggest that the presence of aPL may not be enough to induce a thrombotic event by themselves, being necessary an additional enhancing factor to trigger the thrombosis. De Groot et al. [93] considered the hypothesis of a “second hit”, were the “first hit” would be the presence of the aPL, which promote a procoagulant state, but the thrombotic event would take place when a “second hit” is added to the thrombotic risk. A later study conducted with murine models tested this hypothesis, proving that the thrombotic event only happened when there was an aggression on the vascular wall in the presence of aPL, but not in the presence of the antibodies alone [94].

We remark that the majority of patients in our series did not have a high risk profile of aPL, as 22 of them (82%) had a single positive antibody and most of them (16 patients, 59%) had aCL, predominantly IgG isotype (13 patients). Four patients had two positive aPL, only one patient was triple positive. This observation of a predominant low risk aPL profile supports the concept that an additional risk factor is needed for the thrombosis to happen. This is parallel to the observations in the literature, where 69% of patients had a single positive aPL (most frequently aCL), whereas only 26% were double positive and 5% triple positive. In our series, the presence of aß2GPI was significantly higher compared to the patients previously published. The difference may be because many of the published cases are previous to 2006, when the aß2GPI antibodies were included in the diagnostic criteria in the Sydney criteria revision [17].

Primary APS was the main diagnosis in our series (67%). Sixteen patients (59%) had previous APS manifestations, most of them DVT, which was statistically different from the series of published cases (40%). Only one patient was under treatment with VKA and LDA when the CVT occurred (patient number 2). Interestingly, in our series three patients had had a previous arterial thrombosis: splenic infarctus, radial artery thrombosis and amaurosis fugax. The presence of a concomitant thrombosis was low in our study (7.4%). Seven patients had non-criteria manifestations previous to the CVT or simultaneously: 4 had thrombocytopenia, one had livedo racemosa and two had mitral valvulopathy. Globally this indicates an incidence of 25.9% of non-criteria manifestations, which differs from the 4% found in the literature (only three patients had thrombocytopenia; no other non-criteria manifestations were published). Nevertheless, the patients in our series were admitted to referral centers for APS, which may have been more prone to detect and report the mentioned non-thrombotic manifestations. None of the patients in our series developed catastrophic APS, whereas 3 cases were described in the literature; no significant differences were found between both series.

In our series, only four patients had a new thrombosis (14.8%) during follow-up and none of them was a CVT, which is consistent with previous publications. In the Euro-Phospholipid project 10-year period review of 1000 patients [95], 16.6% of patients developed thrombotic events during the first 5-year period, mainly strokes (5.3%) and transient ischemic attacks (4.7%). In a single center cohort of APS patients followed from 2003 to 2013, the overall rate of recurrent thrombosis was 16.9% despite antithrombotic treatment [96]. Bazzan et al. [97] reported a recurrence rate of 7.5/100 patient years in the first 5 years after the first thrombotic event and identified diabetes, inherited thrombophilia and anticoagulation withdrawal as independent risk factors for thrombosis recurrence. As for CVT, in larger studies the recurrence of CVT was between 2.2% and 4.4% [98] and after discontinuation of anticoagulation, a rate of 0.53 per 100 person-years has been reported [99]. Male gender and the presence of polycythaemia/thrombocythaemia seem to be independent risk factors for a higher risk of recurrence of CVT [100].

During follow-up, the mRS at 6 and 12 months remained favorable, and a disappearance of the remaining symptoms was observed, as many patients with a mRS score of 1 at six months after the CVT (48.1%) were evaluated with a mRS score of 0 at 12 months (59.3%). The mean mRS at six months was 0.79 and at 12 months, 0.52. However, the two patients with a more serious condition due to the CVT had little or no improvement, as one remained with mRS 4 and had several complications over the years after the CVT, and the other one improved to an mRS of 2, with significant impact of the CVT sequelae on the daily basis. This is similar to the data from the ISCVT study [4], where about 80% of the patients made a complete recovery. On the series from the literature, five patients died during the hospital admission for the CVT, one due to catastrophic APS, another had a concomitant stroke (basilar artery), another due to ICHT, another due to infectious complications and in one case there was no specific information. These results are consistent with the predictors of death or dependence identified in the ISCVT study [4]: age over 37 years, male sex, coma, mental status disorder, hemorrhage on admission CT scan, thrombosis of the deep cerebral venous system, central nervous system infection, and cancer.

As for the treatment, the vast majority of patients were treated with anticoagulants (mostly heparin initially, later changed to oral anticoagulants). In our retrospective series, one patient was treated only with corticosteroids (in 1985, and she had CNS bleeding and concomitant thrombocytopenia). During follow-up, VKA was given long-term except in 5 cases, in which VKA was stopped after 6 months due to low risk aPL profile in 3 patients (single positive), two of them treated with LDA. None of those three had recurrences or other thromboses. The patients treated with anticoagulation had a good clinical evolution even though at admission many presented different forms of intracranial bleeding, which is consistent with the ISCVT study [4], where about 40% of patients had a hemorrhagic infarct even before anticoagulant treatment was started. The most recent guidelines from the European Stroke Organization on cerebral venous thrombosis [98] recommend initial treatment with LWMH or UFH when a fast reversal of anticoagulation may be needed (e.g., for neurosurgical intervention); followed by oral anticoagulants (AVK). The authors could not provide a recommendation on thrombolysis for CVT patients, but suggested not using it in patients with low pre-treatment risk of poor outcome [101]. Canhao et al. [102] conducted a systematic review of the literature regarding the use of thrombolytic therapy (both local and systemically administered) in CVT patients, their review suggests that thrombolytics seem to be reasonably safe in these patients. Even so, it is not possible to establish the efficacy of the thrombolytic treatment, so the authors recommend to consign it to critically ill patients non responsive to standard optimal therapy. Siddiqui et al. [103] reviewed the literature for CVT cases treated with mechanical thrombectomy with or without intrasinus thrombolysis, with similar results, as it seems reasonably safe, but there was not enough data to assess efficacy. Endovascular treatment (EVT) of severe CVT was evaluated in a multicenter randomized clinical trial [104], in which EVT along with standard medical care did not appear to be superior to medical care alone. Nonetheless, it was terminated because of futility in a pre-planned interim analyses, so the small sample size does not allow for subgroup analyses that could point to specific benefits (for instance, for patients in a coma due to CVT). In our series of APS patients with CVT, thrombolysis was used only in four patients published in the literature and in none from our retrospective series, which does not allow us to extract any conclusion.

Our study has several limitations due to its retrospective nature, as for many patients we were not able to retrieve any missing data not previously recorded in the medical reports and there is a marked variability in the follow-up time for each patient, so probably we are missing information from the patient’s evolution and, consequently, overestimating the prognosis. As for the review of the literature, a publication bias may have affected the available clinical cases since observations with good results tend to be more easily selected and accepted for publication to the detriment of cases with worse outcome. On that point, we shall remark that the CVT was the first APS manifestation in the majority of our patients and they globally had a good prognosis, perhaps overestimated because more critically ill patients are less frequently published and sometimes the promptness of a rapidly progressing complication stands in the way of a complete clinical research for the underlying cause.

## 5. Conclusions

To our knowledge, our study is the largest and most complete CVT series in APS patients published to date, providing a unique point of view in this rare thrombotic manifestation, that in many cases was the first APS manifestation. When compared to the published CVT cases in APS patients, the clinical picture is homogeneous. The differences found between the two series (arterial thrombosis, non-criteria manifestations, presence of aß2GPI antibody) probably are due to the specialized approach in the three referral hospitals and the fact that many cases from the literature were prior to 2006 (inclusion of aβ_2_GPI as APS criteria).

Our results support the concept that CVT in APS patients is quite similar to CVT by other causes, but they are slightly younger. Interestingly, APS patients presenting with a CVT had an aPL profile of low thrombotic risk and additional CVT risk factors were frequently found, which remarks the hypothesis of a triggering factor that affects a susceptible subject, provoking the final thrombotic event. However, some patients relapsed during follow-up with thrombosis in other locations, which is suggestive of a true underlying thrombophilia. In a patient with CVT, it is essential to search for aPL antibodies, as its presence implies a different therapeutic approach and clinical management in follow-up.

## Figures and Tables

**Table 1 brainsci-11-01641-t001:** Modified Rankin Scale (mRS).

0	No symptoms
1	No significant disability, despite symptoms; able to perform usual duties and activities
2	Slight disability; unable to perform all previous activities but able to look after own affairs without assistance
3	Moderate disability; requires some help, but able to walk without assistance
4	Moderately severe disability; unable to walk without assistance and unable to attend to own bodily needs without assistance
5	Severe disability; bedridden, incontinent, and requires constant nursing care and attention
6	Death

**Table 2 brainsci-11-01641-t002:** CVT characteristics. This table gathers the main information about the characteristics of the CVT, the initial treatment used and the clinical status at discharge, evaluated by mRS. Abbreviations: OC, oral contraceptives; PC deficit, protein C deficit; CS, corticosteroids; PS deficit, protein S deficit; HH, hyperhomocysteinemia; Leiden (het), heterozygous factor V Leiden mutation; NR, not reported; FD, focal deficit; ICHT, intracranial hypertension; AMS, altered mental status; GCS, Glasgow Coma Scale; RTS, right transverse sinus; LTS, left transverse sinus; SSS, superior sagittal sinus; SS, straight sinus; RSS, right sigmoid sinus; LSS, left sigmoid sinus; Jug, jugular vein; CV, cortical veins; VI, venous infarction; PB, parenchymal bleeding; SAH, subarachnoid hemorrhage; VKA, vitamin K antagonists; IVIG, intravenous immunoglobulins; AE, antiepileptic treatment; Acet., acetazolamide.

Patient *n*, Sex, Age		APS Classif.	Risk Factors	Symptoms	Diagnostic Delay	Thrombosed Sinus	CNS Complications	Treatment	mRS
1	F28	SLE	OC	Headache	7 day	RTS	-	VKA	1
2	F31	Primary	-	Headache, FD	16 day	LTS	Cerebellar VI (bilateral)	VKA + antiplatelet	1
3	F29	Primary	Anemia	Headache, FD	30 day	LTS	-	VKA	1
4	F51	Primary	PC deficit, Anemia	Epileptic status-GCS < 9	4 day	LTS, SSS	PB (bilateral), SAH	VKA, IVIG, AE	4
5	F53	SLE	Sinusitis, CS	Headache	6 day	RTS, RSS	-	NR	0
6	M43	ET	Local infection, thrombocytosis	FD, Seizures	NR	NR	NR	VKA	NR
7	F20	SLE/SSc	CS	Headache, ICHT	NR	RTS, LTS, LSS, Jug	-	VKA, Acet.	1
8	F34	Primary	Postpartum	Headache, FD, seizures, ICHT	1 day	RTS, LTS, SSS, SS, Jug	-	VKA, AE	1
9	F44	Primary	After surgery	Headache	NR	RTS, LTS, SSS	SAH (bilateral)	VKA	1
10	F51	Sjögren	Sinusitis	Headache	4 day	CV	PB	VKA, AE	0
11	M33	Primary	-	Headache, FD	3 day	CV	VI	VKA	1
12	F24	Primary	Sinusitis	Headache	1 day	LTS, LSS, Jug	-	VKA	1
13	F23	Primary	-	Headache	30 day	RTS, RSS, Jug	-	VKA, Acet.	1
14	F41	SLE	OC	Headache	15 day	LTS	Cerebellar VI	VKA	1
15	F43	Primary	OC, PS deficit	Headache	8 day	LTS, SSS	SAH	VKA	1
16	M23	SLE	-	Headache, ICHT	48 day	RTS, SSS, Jug, CV	-	VKA	1
17	F20	Primary	OC, HH	Headache, FD, Seizures, AMS	NR	RTS, LTS, SSS, SS	PB (bilateral)	VKA	1
18	F70	ET	-	Headache, FD, AMS	7 day	LTS, SSS, SS	VI (bilateral)	VKA	1
19	F39	Primary	OC	Headache, ICHT	3 day	LTS, LSS, Jug	-	VKA	0
20	F40	Primary	OC, local infection	Headache, Seizures, AMS	1 day	SSS	Bleeding infarction (bilateral)	VKA	1
21	F16	Primary	Local infection	Headache	7 day	LTS	-	VKA	0
22	F23	Primary	OC	Headache	NR	RTS, SSS	-	VKA	0
23	F32	Primary	-	Headache, AMS	5 day	LTS, Jug	PB, SAH	VKA, IVIG	1
24	M36	SLE	-	Headache	1185 day	RSS	-	VKA	1
25	F16	Primary	-	Headache, AMS	10 day	LTS	Bleeding infarction	CS	1
26	F38	Primary	OC, Leiden (het)	Headache, FD	3 day	LTS, LSS, Jug	PB	VKA	0
27	M64	Primary	-	Headache, Seizures, AMS-GCS < 9	3 day	SSS	PB (bilateral)	VKA	4

**Table 3 brainsci-11-01641-t003:** APS characteristics. This table gathers the main information about the APS characteristics. It is noted whether it was primary APS or associated with another disease, as well as the criteria set used at diagnosis. Based on the antibodies, patients were classified into the following categories: I, more than one laboratory criteria present (any combination); II_a_, LA present alone; II_b_, aCL present alone; II_c_, anti-β_2_-glycoprotein-I present alone. Abbreviations: F, female; M, male; SLE, systemic lupus erythematosus; ET, essential thrombocytosis; SSc, systemic sclerosis; DVT, deep vein thrombosis; UL, upper limbs; LL, lower limbs; PE, pulmonary embolism.

Patient *n*, Sex, Age		APS Classif.	APS Criteria	APS Antibodies	CVT as APS First Manifestation	Concomitant Thromboses	Previous Thrombotic Manifestations	Previous Obstetrical Manifestations	Non-Criteria Manifestations
1	F28	SLE	Sapporo	I-LA, aCL IgG	Yes	-	Amaurosis fugax	-	Mitral valvulopathy
2	F31	Primary	Sapporo	II_b_-IgG	No	-	Splenic infarction	Stillbirth (24 weeks), placental insufficiency 2 miscarriages	-
3	F29	Primary	Sydney	II_c_	Yes	-	-	-	-
4	F51	Primary	Sapporo	II_b_-IgG	Yes	-	-	-	Thrombocytopenia
5	F53	SLE	Sapporo	II_b-_IgM, IgG (shift)	No	-	DVT (UL + LL), PE Radial artery thrombosis	-	Mitral valvulopathy
6	M43	ET	Sapporo	II_b_-IgG	Yes	-	DVT (LL, asymptomatic)	-	-
7	F20	SLE/SSc	Sydney	II_b_-IgG	Yes	-	-	-	-
8	F34	Primary	Sydney	II_b_-IgG	Yes	-	PE	-	-
9	F44	Primary	Sydney	II_b_-IgG	No	-		One miscarriage	-
10	F51	Sjögren	Sydney	II_c_	Yes	-	-	-	-
11	M33	Primary	Sydney	II_b_-IgG	Yes	DVT (LL)	-	-	-
12	F24	Primary	Sydney	II_c_	No	-	3 DVT (LL)	-	-
13	F23	Primary	Sydney	II_b_-IgG	No	-		2 miscarriages, 2 stillbirths (20 weeks)	Livedo racemosa
14	F41	SLE	Sapporo	II_b_-IgM, IgG	No	-	DVT (UL) + PE	1 miscarriage	-
15	F43	Primary	Sydney	II_b_-IgM	Yes	-	-	-	-
16	M23	SLE	Sydney	II_b_-IgG	Yes	-	-	-	-
17	F20	Primary	Sydney	II_b_-IgG	Yes	-	-	-	-
18	F70	ET	Sydney	II_b_-IgM	Yes	-	-	-	-
19	F39	Primary	Sydney	I-LA, aCL, aβ_2_GPI	Yes	-		One miscarriage	-
20	F40	Primary	Sapporo	II_b_-IgM	Yes	-	-	-	-
21	F16	Primary	Sydney	I-aCL, aβ_2_GPI	Yes	-	-	-	Thrombocytopenia
22	F23	Primary	Sydney	II_b_-IgG	Yes	-	-	-	-
23	F32	Primary	Sydney	II_a_	Yes	-	-	-	Thrombocytopenia
24	M36	SLE	Sydney	II_a_	No	-	PE	-	-
25	F16	Primary	Sapporo	I-LA, aCL	Yes	-	-	-	Thrombocytopenia
26	F38	Primary	Sydney	I-LA, aCL	Yes	-	-	-	-
27	M64	Primary	Sapporo	II_a_	Yes	PE	-	-	-

**Table 4 brainsci-11-01641-t004:** Evolution of CVT patients during follow up. This table gathers the main information about the evolution of CVT patients during follow up. Abbreviations: NR, not reported; CS, corticosteroids; SLE, systemic lupus erythematosus; ET, essential thrombocytosis; SSc, systemic sclerosis; DVT, deep vein thrombosis; UL, upper limbs; LL, lower limbs; PE, pulmonary embolism; CVA, cerebrovascular accident; mSR, modified Rankin Scale.

Patient *n*, Sex, Age		APS Classif.	Months of Follow-Up	INR	Treatment Duration	Neuroimaging	Thrombosis and Hemorrhages	Sequelae	mRS 6 Months	mRS 12 Months
1	F28	SLE	108	3–3.5	Long-term	Partial reperm.	DVT (LL), TIA. Psoas hematoma	-	1	0
2	F31	Primary	82	3–3.5	Long-term	Complete reperm.	LL hematoma	Headache	1	0
3	F29	Primary	38	3–3.5	Long-term	NR	-	Headache	1	1
4	F51	Primary	91	3–3.5	6 years	Partial reperm.	-	Seizures	2	2
5	F53	SLE	20	3–3.5	Long-term	Persistent thrombosis	-	-	1	0
6	M43	ET	289	NR	5 months	NR	-	Seizures	0	0
7	F20	SLE/SSc	45	3–3.5	Long-term	Partial reperm.	-	Headache	1	0
8	F34	Primary	78	2.5–3	Long-term	Complete reperm.	-	Headache	1	1
9	F44	Primary	53	2–3	Long-term	NR	-	NR	NR	NR
10	F51	Sjögren	54	NR	7 months	Persistent thrombosis	-	-	0	0
11	M33	Primary	12	2–3.5	Long-term	Partial reperm.	-	-	0	0
12	F24	Primary	26	2–3	Long-term	Partial reperm.	-	-	0	0
13	F23	Primary	13	2–3	Long-term	Partial reperm.	-	Headache	1	1
14	F41	SLE	1	2.5–3.5	Long-term	NR	CVA (Sylvian). SAH	Seizures, cognitive disturbances	NR	1
15	F43	Primary	37	2–3	Long-term	Complete reperm.	LL hematoma	Headache	1	0
16	M23	SLE	19	2–3	Long-term	Partial reperm.	-	Headache, seizures	1	0
17	F20	Primary	24	2.5–	Long-term	Partial reperm.	-	Headache, seizures	1	1
18	F70	ET	48	2–3	Long-term	Complete reperm.	-	-	1	0
19	F39	Primary	91	2–3	6 months	Partial reperm.	-	-	0	0
20	F40	Primary	146	2–3	Long-term	Complete reperm.	-	-	0	0
21	F16	Primary	1	3–3.5	Long-term	NR	-	-	NR	NR
22	F23	Primary	117	NR	6 months	Partial reperm.	-	Dural fistula type I	0	0
23	F32	Primary	17	2–3	Long-term	Partial reperm.	Parenchymal hematoma	Headache, cognitive disturbances	1	1
24	M36	SLE	7	2–3	Long-term	NR	-	-	0	0
25	F16	Primary	170	(CS)	(CS)	Persistent thrombosis	DVT (LL)	Headache	1	1
26	F38	Primary	109	2–3	Long-term	Complete reperm.	-	-	0	0
27	M64	Primary	90	2–3	6 months	NR	DVT (LL, 4 times), PE	Focal deficit, seizures	4	4

**Table 5 brainsci-11-01641-t005:** Cases of patients with CVT and APS published in the literature: clinical characteristics and outcome. * mRS at hospital discharge. Abbreviations: F, female; M, male; APS, antiphospholipid syndrome; SLE, systemic lupus erythematosus; ET, essential thrombocytosis; SSc, systemic sclerosis; LA-HPS, Lupus anticoagulant-hypoprothrombinemia syndrome; CON, cryptogenic organising pneumopathy; UCTD, undifferentiated connective tissue disease; DVT, deep vein thrombosis; UL, upper limbs; LL, lower limbs; PE, pulmonary embolism; Thrombocytop, thrombocytopenia; LP, lumbar puncture; OC, oral contraceptives; PC deficit, protein C deficit; CS, corticosteroids; PS deficit, protein S deficit; HH, hyperhomocysteinemia; Leiden (het), heterozygous factor V Leiden mutation; Leiden (hom), homozygous factor V Leiden mutation; LOC, loss of consciousness; FD, focal deficit; ICHT, intracranial hypertension; AMS, altered mental status; GCS, Glasgow Coma Scale; LA, lupus anticoagulant; aCL, anticardiolipin antibodies; aβ_2_GPI, anti-β_2_-glycoprotein-I antibodies; RTS, right transverse sinus; LTS, left transverse sinus; TS, transverse sinus; SSS, superior sagittal sinus; SS, straight sinus; RSS, right sigmoid sinus; LSS, left sigmoid sinus; Jug, jugular vein; CV, cortical veins; VI, venous infarction; PB, parenchymal bleeding; SAH, subarachnoid hemorrhage; VKA, vitamin K antagonists; IVIG, intravenous immunoglobulins; AE, antiepileptic treatment; Therap. LMWH, therapeutic low molecular weight heparin; Therap. UFH, therapeutic unfractioned heparin; Acet., acetazolamide; mSR, modified Rankin Scale; NR, not reported.

Authors	Sex, Age	Diagnosis	Previous APS Manifestations	Risk Factors for CVT	CVT Symptoms	APS Antibodies	Thrombosis	CNS Complications	Treatment	mRS *
Feki et al. [24]	F36	APS LA-HPS	Miscarriages	factor II (37%).	Headache (2 weeks), FD, seizure, transient LOC	LA, aCL IgG	SSS, CV	Subdural hemorrhage	CS, AE, heparin, VKA	NR
Shlebak [25]	F(20–29)	APS	PE (2)	OC, obesity smoker	Headache	LA	TS	-	Therap. LMWH	0
	F(20–29)	APS	-	OC	Headache (1 week)	LA, aCL IgG, aβ_2_GPI IgG	TS, Sigmoid sinus	-	Therap. LMWH, then VKA	0
	M(30–39)	APS	-	Smoker	Headache, ICHT	LA, aβ_2_GPI IgG	TS	-	Therap. LMWH, then VKA	0
Ho et al. [26]	F50	Secondary APS (Sjögren)	-	-	Headache (1 day)	LA	LTS	-	Therap. LMWH, then VKA	NR
Mahale et al. [27]	F25	APS	Miscarriages (2)	Pregnancy (miscarriage 5 day prior CVT)	Headache, ICHT, Seizures, GCS < 9	LA, aCL IgG and IgM	LTS, LSS, Jug	Vertebral and basilar artery thrombosis	Heparin, antiplatelet, mannitol	6
Behrendt et al. [28]	F35	APS	-	Sinusitis, mastoiditis, contraception ring	Headache (3 weeks), ICHT, FD (abducens palsy), AMS	LA, aCL IgG and IgM, aβ_2_GPI IgG and IgM	SSS, RTS	-	NFH, then VKA Acetaz., ventriculo-peritoneal shunt	NR
Pelegrina Molina et al. [29]	F16	APS	-	-	Headache (1 week), ICHT	LA, aCL IgG, aβ_2_GPI IgG	RTS, RSS	-	CS, VKA, Acet.	NR
Tsai et al. [30]	F49	APS	-	-	Repeated TIA (1 day)	LA	SSS	Vasogenic edema	Therap. LMWH, then VKA	0
Hua et al. [31]	F25	CAPS	-	-	Headache, neck rigidity	LA, aCL	SSS	-	CS, mannitol, IVIG, LMWH	6
Hanprasertpong et al. [32]	F20	APS	-	Pregnancy (10 weeks)	Headache, FD (left hemiparesis)	LA	RTS, RSS, CV	-	Therap. LMWH (pregnancy), then VKA	0
Sakamoto et al. [33]	M38	APS	-	-	Headache (8 weeks), ICHT, FD (abducens palsy)	LA, aCL	SSS, TS, Sigmoid sinus, Jug	-	Therap. UFH, then VKA. Mannitol, Ventriculo-peritoneal shunt	NR
Pendse et al. [34]	F19	CAPS	-	Obesity (BMI 42.3), multiple LP	Headache (16 weeks)	LA	SSS, RTS	-	Anticoagulation	NR
Bonnet et al. [35]	M26	Secondary APS (Behçet)	-	-	Headache	LA	SSS, TS	-	Therap. LMWH, then VKA. CS	0
Arunkalaivanan et al. [36]	F32	APS	HELLP	Postpartum	Seizures, GCS < 9	aCL IgM	SSS	VI	Therap. UFH, Therap. LMWH, then VKA. AE.	1
Yuen et al. [37]	F14	Secondary APS (SLE)	-	-	Headache (4 weeks)	aCL	RTS, RSS, Jug	-	Therap. LMWH, then VKA. CS	0
Viswanathan et al. [38]	M30	Sneddon Syndrome	-	Smoker	Seizures, GCS < 9	LA, aCL IgG and IgM	SSS, SS	-	Therap. heparin, then VKA. Aspirin, CS	NR
Nagai et al. [39]	F36	APS	Miscarriages (2), Stillbirth	Postpartum (9 day, premature)	AMS progresses to GCS < 9, ICHT, FD (right hemiparesis, aphasia).	LA, aCL IgG and IgM, aβ_2_GPI	SSS, CV	-	Mannitol, AE, CS. External decompression	NR
Kesler et al. [40]	F35	APS	Miscarriages (3)	-	Headache, ICHT	LA, aCL IgG and IgM	SSS, RSS	-	Therap. UFH, Therap. LMWH, CS.	0
Levine et al. [41]	F21	APS	Miscarriage, DVT (LL), PE Thrombocytop	Pregnancy	Headache, ICHT, FD	LA	TS	-	Therap. LMWH, then VKA. CS	NR
	M32	APS	Thrombocytop	-	Headache, ICHT, FD	LA	SSS, SS	-	Therap. heparin, then VKA. CS	NR
Moreb et al. [42]	F20	APS	-	Postpartum	Seizures, FD	LA	NR	VI	Therap. LMWH, then VKA. AE	1
Vidailhet et al. [43]	F24	Secondary APS (SLE)	Stillbirth	Postpartum	Headache, Seizures, FD	aCL	SSS, CV	PB	Therap. LMWH, then VKA. CS	1
	F31	Secondary APS (SLE)	DVT (LL)	-	Headache, Seizures, FD	LA	SSS, TS	VI	Therap. LMWH, then VKA. CS	1
	F52	Secondary APS (SLE)	-	-	Headache, Seizures	LA	TS, Jug	NR	Therap. LMWH, then VKA. CS	1
Mokri et al. [44]	F49	APS	-	-	ICHT, Headache, vision impairment	aCL IgG	RTS, LTS	VI	VKA. Acet.	1
Provenzale et al. [45]	M62	APS	-	-	Headache, ICHT, Seizures, FD	LA	SSS, LTS, LSS	PB, VI	Terap. heparin, then VKA	3
	M25	APS	-	-	Headache, ICHT, Seizures, FD	LA	SSS, RTS	PB, VI	Terap. heparin, then VKA. Acet.	NR
Boggild et al. [46]	M30	Secondary APS (Behçet)	DVT (LL)	-	Headache, ICHT	LA	SSS	-	Anticoagulation. Acet. CS	0
	F42	APS	-	After surgery	Headache, ICHT	LA	LSS	-	Antiplatelet	0
Camaiti et al. [47]	F49	APS	-	-	Headache, Seizures	aCL	LTS, RTS, Jug	PB	Therap. heparin, then VKA	NR
Khoo et al. [48]	M51	APS	-	-	Headache (2 day), ICHT, FD, AMS	LA	SSS, LTS, RTS, SS	Cerebral edema	Therap. heparin, Thrombolysis, VKA, CS	0
Agah et al. [49]	F44	APS	-	After surgery, hormonal therapy	Headache, ICHT, Seizures, FD, GCS < 9	LA, aCL IgG	SSS, LTS, RTS, CV	PB, VI	Therap. heparin, mannitol, AE	6
Silburn et al. [50]	F50	APS	DVT (LL), PE	Hormonal therapy (6 weeks)	Headache (5 days), AMS, ICHT	aCL	SSS, LTS, SS	VI (bilateral)	Anticoagulation	3
	F26	APS	-	OC	Headache, Seizures, FD (aphasia), AMS	aCL	SSS, RTS, LTS, SS	VI (bilateral)	Anticoagulation	4
Ricchieri et al. [51]	F30	APS	-	Transitory PS deficit	Seizures, FD (hemiparesis), AMS	aCL IgG	SSS	PB, VI	Therap. heparin, then VKA	0
Hillier et al. [52]	F27	APS	NR	PC deficit	Headache, ICHT	LA	NR	NR	Therap. heparin, then VKA. Ventriculo-peritoneal shunt	2
	F36	APS	NR	-	Headache, Seizures	LA	NR	NR	Mannitol, CS	6
Deschiens et al. [53]	F25	Secondary APS (SLE)	-	Leiden (het) Postpartum	Headache, Seizures, FD	LA, aCL	SSS, LTS	PB	Therap. heparin, then VKA.	1
	F30	APS	Miscarriage, DVT	Leiden (hom)	Headache, FD (aphasia)	aCL	SSS, RTS	-	Therap. heparin, then VKA.	NR
	F23	APS	-	OC	Headache, ICHT	aCL	SSS, RTS	-	Therap. heparin, then VKA.	NR
Carhuapoma et al. [54]	F23	NR	NR	Postpartum, OC	Seizures	aCL	LTS, CV	NR	Thrombolysis	NR
	F30	APS	NR	OC	Headache, FD	aCL	SSS, LTS, CV	-	Antiplatelet	NR
	F22	APS	-	Postpartum, OC	Headache, FD, AMS	aCL	LTS, CV	-	Therap. heparin, then VKA.	NR
	F36	APS	Miscarriage	Postpartum	Headache	aCL	SSS, LTS	-	Thrombolysis. Therap. heparin, then VKA.	NR
	M19	APS	-	-	Headache, FD	aCL	SSS, LTS, CV	-	Therap. heparin, then VKA.	NR
	F33	APS	-	Postpartum	Headache	aCL	RTS, Jug	-	Therap. heparin, then VKA.	NR
	F24	APS	Miscarriage	Postpartum	Headache, Seizures, AMS	aCL	SSS	-	VKA	NR
Veilhaber et al. [55]	F13	APS	DVT	Otitis	Headache	aCL	LTS	-	NR	NR
	F16	APS	NR	Type I PC	Headache	aCL IgG	SSS	-	NR	NR
Kim et al. [56]	F25	APS	NR	-	FD	aCL	SSS, RTS, RSS	PB, VI	NR	NR
Dzialo et al. [57]	M34	APS	-	Epidural infiltration	Headache, ICHT, Seizures, FD	LA	SSS	PB, VI	Thrombolysis. Therap. heparin, then VKA. AE. Mannitol	3
Marietta et al. [58]	F19	APS	DVT (LL), PE	-	Headache	LA, aCL IgG	SSS	-	Therap. LMWH, then VKA	0
Kao et al. [59]	F20	Crohn D.	-	-	Headache	aCL IgG	SSS, RTS, LTS	VI	Therap. LMWH, then VKA	1
Madhavan et al. [60]	F25	APS	Miscarriage	Pregnancy	ICHT, FD	LA, aCL IgG and IgM	NR	VI	Therap. LMWH, then VKA	NR
Sareen et al. [61]	F17	APS	-	Anemia	ICHT	aCL	SSS, RTS	-	Therap. heparin, then VKA. CS	0
	M20	APS	-	-	ICHT	LA, aCL	SSS	-	Tharap. heparin, then VKA	0
	F25	APS	-	Postpartum	ICHT	aCL	SSS, SS	VI (bilateral)	Therap. heparin, then VKA. CS	0
Pagnoux et al. [62]	M68	APS, Sweet S.	Portal thrombosis	Thalidomide, myelodysplastic syndrome	FD (left hemiplegia)	LA, aCL IgG	SSS	VI	Therap. heparin, then VKA	1
Roldan-Molina et al. [63]	F13	Secondary APS (SLE)	-	Leiden	Headache, ICHT	LA	RTS, RSS, Jug	-	Therap. heparin, then VKA. CS	NR
Uthman et al. [64]	F18	Secondary APS (SLE)	-	Leiden (het)	Headache	LA	RSS, Jug	-	Anticoagulation	NR
Cakmak et al. [65]	M22	APS	-	-	Headache, Seizures	LA, aCL IgG	RTS	PB, VI	Therap. heparin, then VKA	1
Vujovic et al. [66]	F30	APS	Miscarriage (7), premature birth (1)	Factor II mut. Mastoiditis, Obesity	NR	LA	RTS	-	Therap. heparin, then VKA. Acet.	NR
Appenzeller et al. [67]	F23	APS	NR	-	Seizures, AMS	NR	TS	-	Therap. heparin, then VKA	0
	F19	Secondary APS (SLE)	NR	-	Headache, ICHT, FD	NR	TS, SS	VI (bilateral)	Therap. heparin, then VKA	2
	F35	APS	NR	-	Headache, ICHT, Seizures, FD	NR	SSS	VI	Therap. heparin, then VKA	1
Mutukhumar et al. [68]	M38	APS	-	PC deficit. Cranial traumatism	ICHT, FD (abducens palsy)	NR	SSS	-	VKA. Acet.	1
Amberger et al. [69]	F31	APS	CVT	Increased factor VIII, OC	Headache	aCL	SSS, RTS	-	Therap. heparin, then VKA.	0
Lega et al. [70]	M20	APS	DVT (LL, 2)	After surgery Leiden	ICHT, FD	LA	SSS, RTS	VI	Therap. heparin	NR
Zeller et al. [71]	F20	CAPS	DVT (LL)	-	Seizures, FD	LA, aCL IgM	NR	NR	Therap. LMWH, VKA. CS, AE, plasmapheresis	NR
Varma et al. [72]	M26	APS	-	Anemia, Nephrotic Syndrome	Seizures	AL, aCL	TS, RSS, LSS	PB (bilateral)	VKA.	NR
Miki et al. [73]	M38	APS	-	Dural fistula, Cranial traumatism	AMS (altered behavior)	β_2_GPI	SS	VI (bilateral)	Anticoagulation, Fistula embolization, surgery	NR
Chen et al. [74]	M21	APS	-	Anemia	Headache, ICHT	LA	SSS, LTS, LSS, Jug	-	Antiplatelet, CS, ciclosporin A	0
Panda et al. [75]	M25	APS	NR	-	Headache, ICHT, Seizures, FD	NR	SSS	PB, VI, SAH	Therap. heparin, then VKA.	2
Hajialilo et al. [76]	F20	Secondary APS (SLE)	-	Pregnancy	Headache, ICHT, Seizures	LA, aCL IgG	RTS, RSS	-	Thrombolysis. Therap. heparin, then VKA. CS	0
Demir et al. [77]	F31	Secondary APS (SLE)	NR	Pregnancy	Headache, ICHT, FD (III palsy)	NR	SSS, TS	-	Therap. heparin, then VKA.	0
	F22	APS	NR	Pregnancy	Headache, ICHT, FD (III palsy)	NR	SSS, SS	VI	Therap. LMWH, then VKA.	1
Ikejiri et al. [78]	F18	APS	NR	-	NR	LA, aβ_2_GPI IgG	RTS	NR	NR	NR
Butala et al. [79]	F37	Secondary APS (LES)	DVT	Cranial traumatism	Headache, AMS	NR	RTS, LTS, RSS, LSS	-	Therap. heparin	NR
Numata et al. [80]	M45	APS	-	Tadalafil	Headache (6 day), Seizures, AMS	aCL, aβ_2_GPI	CV	VI	Therap. heparin, AE	0
Hsieh et al. [81]	F67	APS, CON	-	Hormonal therapy	Headache	aβ_2_GPI IgM	RTS, RSS, Jug	-	Therap. UFH, Therap. LMWH, VKA. CS	NR
Castillo et al. [82]	F19	APS	-	OC, Otitis	Headache (2 weeks), ICHT, FD (abducens palsy)	LA, aCL	SSS, LTS, RTS	-	Therap. heparin, then VKA	0
Keller et al. [83]	F36	APS	HELLP, DVT	Postpartum, after surgery	Headache (4 weeks), ICHT, FD	aCL	SSS	Ictus	Therap. LMWH, then VKA.	3-4
Nyo et al. [84]	F26	Secondary APS (UCTD)	HELLP, Thrombocytop	Anemia	Headache	LA	NR	Cerebral edema	HBPM, CS	6
Polster et al. [85]	F16	APS	Preeclampsia	Postpartum	FD (dysarthria, right VII palsy and hypoesthesia), Seizures, ICHT	LA	CV	Unstable PB, Cerebral edema	Decompressive craniectomy, AE, Mannitol, external ventricular drain	4
Elnahry et al. [86]	M21	APS	DVT (LL)	HH	Headache (4–5 weeks), vision impairment, ICHT, AMS (confusion), fever	LA	SSS, LTS	-	Therap. LMWH, VKA, Acet.	1

**Table 6 brainsci-11-01641-t006:** Clinical characteristics of the CVT in the literature compared to our series. The last column describes the characteristics of the APS patients presenting with a CVT (both from the literature and our series). Abbreviations: CVT, cerebral vein thrombosis; APS, antiphospholipid syndrome; ICHT, intracranial hypertension; AMS, altered mental status; GCS, Glasgow Coma Scale; mRS, modified Rankin Scale; LA, lupus anticoagulant; aCL, anticardiolipin antibodies; aβ_2_GPI, anti-β_2_-glycoprotein-I antibodies.

Variables	Retrospective Series (*n* = 27)	Series from the Literature (*n* = 86)	*p*	Total CVT Cases in APS Patients (*n* = 113)
Women	22/27 (82%)	62/86 (72%)	0.41	85/113 (75%)
Age	35.7 years (16–70)	30.5 years (13–68)	0.07	31.8 years (13–70)
Clinical presentation				
acute	3/22 (14%)	31/62 (50%)	0.002	34/84 (40%)
subacute	15/22 (68%)	26/62 (42%)	0.08	41/84 (49%)
chronic	4/22 (18%)	5/62 (8%)	0.11	9/84 (11%)
CVT risk factors				
additional risk factors	18/27 (67%)	55/86 (64%)	0.99	73/113 (64%)
pregnancy/postpartum	1/18 (6%)	20/57 (35%)	0.01	21/75 (28%)
oral contraceptives	8/18 (44%)	10/63 (16%)	0.01	18/81 (22%)
other thrombophilia	4/27 (15%)	13/86 (15%)	0.93	18/113 (16%)
local infections	6/27 (22%)	4/86 (5%)	0.005	10/113 (9%)
Symptoms				
headache	24/27 (89%)	64/86 (74%)	0.12	88/113 (78%)
ICHT	3/10 (30%)	37/86 (43%)	0.47	40/96 (42%)
focal deficit	8/27 (30%)	37/86 (44%)	0.22	45/113 (40%)
seizures	6/27 (22%)	28/86 (33%)	0.32	34/113 (30%)
AMS/GCS < 9	7/27 (26%)	18/86 (21%)	0.53	25/113 (22%)
Cvt location				
transverse and sigmoid sinus	22/26 (85%)	55/80 (69%)	0.12	77/106 (73%)
superior sagittal sinus	10/26 (39%)	48/80 (60%)	0.05	58/106 (55%)
straight sinus and deep venous cerebral system	3/26 (12%)	8/80 (10%)	0.85	11/106 (10%)
cortical veins	3/26 (12%)	11/80 (14%)	0.86	14/106 (13%)
extensive thrombosis	15/26 (58%)	48/80 (60%)	0.87	62/106 (58%)
mRS at discharge				
mRS 0–1	24/27 (89%)	37/51 (73%)	0.11	61/78 (78%)
mRS 2	0/27	3/51 (6%)	0.6	3/78 (4%)
mRS 3–5	2/27 (7%)	6/51 (11%)	0.68	8/78 (10%)
mRS 6	0/27	5/51 (10%)	0.27	5/78 (6%)
Primary APS	18/27 (67%)	67/85 (79%)	0.07	85/112 (76%)
APS-SLE	6/27 (22%)	11/85 (13%)	0.26	17/112 (15%)
APS-other causes	3/27 (11%)	6/85 (7%)	0.52	9/112 (8%)
Previous APS manifestations				
total	16/27 (59%)	28/70 (40%)	0.02	44/97 (45%)
venous thromboses	6/27 (22%)	15/70 (21%)	0.48	21/97 (22%)
arterial thromboses	3/27 (11%)	0/70	0.03	3/97 (3%)
obstetric manifestations	5/18 (28%)	15/46 (32%)	0.92	20/64 (31%)
non-criteria manifestations	9/27 (33%)	3/70 (4%)	<0.001	12/97 (12%)
Catastrophic APS	0/27	3/85 (3.6%)	0.99	3/112 (2.7%)
Immunologic characteristics				
aCL	19/27 (70%)	51/78 (66%)	0.75	70/105 (67%)
LA	7/27 (26%)	44/78 (56%)	0.008	51/105 (49%)
aβ_2_GPI	5/27 (19%)	7/78 (9%)	0.03	12/105 (11%)
single positive	22/27 (82%)	54/78 (69%)	0.19	76/105 (72%)
double positive	4/27 (15%)	20/78 (26%)	0.19	24/105 (23%)
triple positive	1/27 (4%)	4/78 (5%)	0.22	5/105 (5%)

## Data Availability

Data available on request due to privacy restrictions.

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
