# Peer review of "Cerebral Vein Thrombosis in the Antiphospholipid Syndrome: Analysis of a Series of 27 Patients and Review of the Literature"

_brainsci, 2021, doi:10.3390/brainsci11121641_

Round 1

Reviewer 1 Report

This is really interesting work. In it, the authors try to make a more systematic study of a phenomenon as rare as cerebral vein thrombosis in APS.

1. On page 6:

- First paragraph  “To  diagnose  CVT,  a  combination  of  imaging  techniques  was  performed in most patients (12 out of 25)” authors must change “most patients” to “half of patients”.

- Second paragraph:  “Another risk factor for CVT was identified in 18 patients (66.7%); the most frequent one was the use of oral contraceptives (8 patients), followed by local infections (6 patients). Eighteen patients (63%) had at least one additional risk factor besides APS and seven patients (25.9%) had two or more risk factors besides APS”.  

Authors should review all percentages, since they give 2 different percentages (66,7% and 63%) with respect to the same number of patients(18).

2. If "sixteen patients (59%) had previous  APS  manifestations" as authors say. At the moment that CVT occurred, was there any patient under anticoagulant therapy? If affirmative, it should be added to text.

3. It would be very interesting to know the delay time between the thrombotic event and the determination of the antiphospholipid antibodies, if it is not possible to know it in each case, it would be useful to know the median time.

4. In table 3, please explain/correct the acronym “SAPL”

5. As authors discuss, main aPL profile was aCL IgG, was there any difference in the extension or severity,  sequels  etc… as well as in the follow-up with the rest of the aPL profiles?.

Reviewer 2 Report

the paper is interesting as depicts a large series of APS patients with cerebral vein thrombosis.

is there any difference in terms of autoantibody isotype?

Could you detect any change in the autoantibody profile in patients who subsequently had cerebral vein thrombosis compared with aPL profile at disease onset?

please change B2GPI in the tables and in the text with aB2GPI (when referring to the autoantibody).

was there any patient in which HLA (including HLAB51) was tested?

which was the prevalence of other manifestations including thrombocytopenia, livedo, aphtous ulcers?
